# [Re] Domain Generalization using Causal Matching

## Reproducibility Summary

**Scope of Reproducibility**

We reproduced the results of the paper "Domain Generalization Using Causal Matching." Traditional supervised learning assumes that the classes/labels seen in testing must have appeared during the training phase. However, this assumption is often violated in real-world applications. For instance, in e-commerce, new categories of products are released every day. A model that cannot detect new/unseen classes is hard to function in such open environments as they are not generalizable.

**Methodology**

The open-source code of the paper has been used. The authors provided detailed instructions to reproduce the results on their GitHub page. We reproduced almost every table in the main text and few of them from the appendix. In case of a mismatch of the results, we also investigated the cause and proposed possible explanations for such behavior. For the extensions, we wrote extra functions to check the paper's claim on other open-source standard datasets. We mainly used the infrastructure offered by the publicly available GPUs offered by Google Colab and GPU-assisted desktop computers to train the models.

**Results**

Most of our results closely match the reported results in the original paper for the Rotated-MNIST [17], Fashion-MNIST [27], PACS [18, 28], and Chest-Xray [3] datasets. However, in some cases, as described later, we obtained better results quantitatively than the ones reported in the paper. By investigating the root cause of such mismatches, we provide a possible reason to avoid such a gap. We performed additional experiments by making necessary modifications for the Rotated-MNIST and Rotated Fashion-MNIST dataset. In general, our results still support the main claim of the original paper, even though the results differ for some of the training/testing instances.

**What was easy**

The official GitHub page of the paper has the open-source code, which was beneficial as it was well organized into multiple files. Thus, it was easy to follow. The experiments described in the paper were done on widely-used standard open-source datasets. Therefore, implementing each experiment was relatively easy to do. Furthermore, since most of the parameters were reported in the scripts, we did not need much tuning in most experiments.

**What was difficult**

Though implementing each experiment is relatively simple, the numerosity of experiments was a hard task. In particular, each experiment in the original setting requires training a network for a significant number of iterations. Having limited access to computational resources and time, we sometimes changed the settings, sacrificing granularity. However, these changes did not affect the interpretability of the final results.

**Communication with original authors**

We emailed the authors and received prompt responses to our questions regarding the provided Jupyter reproduction notebooks. Some tables had multiple runs for the same technique, but it was unclear how to execute the alternative runs.

Preprint. Under review.

# 1   Introduction

Learning is a dynamic process in an open environment where some new classes may not belong to any training set; therefore, identifying these novel classes during classification presents an important problem. The objective of domain generalization is to learn a single classifier with training data sampled from M domains that generalize well to data from unseen domains. For example, a model trained on certain attributes of one region may be deployed to another, or an image classifier may be deployed on slightly rotated images. This proposition assumes that stable (causal) features lead to an optimal classifier invariant to the domains.

The paper illustrates that the class-conditional domain invariant objective for representations is not always sufficient. They provide simple counterexamples to validate the class-conditional domain invariance insufficiency theoretically and empirically. Differing distributions of stable causal features within the same class label are commonly observed in real-world datasets, e.g., in digit recognition, the stable feature like shape may differ based on people's handwriting, or medical images may have variations due to differing body characteristics in the sample. The paper proposes the importance of considering within-class variation in stable features.

This report repeats the original paper's experiments and compares them with the reported results. Also, we extend the original paper results by investigating the effect of data augmentation on Rotated-MNIST and Rotated Fashion-MNIST datasets under different settings. We report and discuss our results in later sections.

Domain generalization is a phenomenon that can generalize to unseen data distributions after training on more than one data distribution. For example, a model trained on one domain may be deployed to another, i.e., domain adaptability or an image classifier may be deployed on slightly rotated images. The goal is to "learn representations independent of the domain after conditioning on the class label" [21].

The paper investigates the observation through a structural causal model (SCM) and discusses the importance of modeling within-class variations for generalization. The authors [21] propose new methods RandMatch, MatchDG, and MDGHybrid to increase performance over the previous state-of-the-art methods for various ML problems. In addition to reproducing the original paper's results, we propose different state-of-the-art datasets where the analogy can be implemented and assess the effectiveness of the proposition.

# 2   Scope of reproducibility

The paper broadly tackles the issue of spurious correlation where some predictive attributes in the training time might not be predictive at the test time. For example, in Figure 1, we can observe the two different domains on which a cow could appear in and/or be trained. If a learning algorithm does not use domain-independent features and has most if not all training images of an object in one domain, it may fail when attempting to identify in other domains.

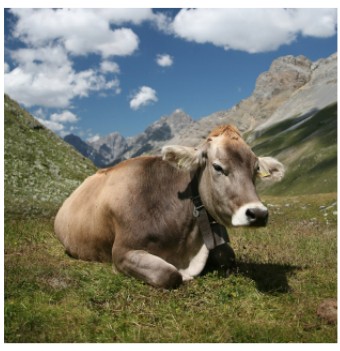 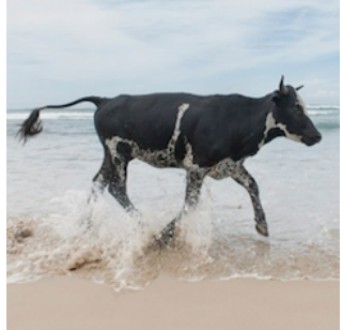 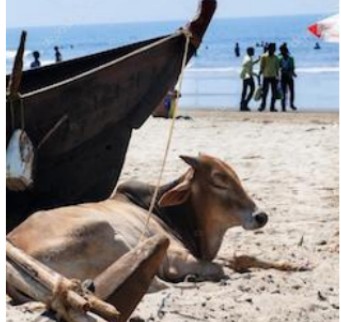

(A) **Cow: 0.99**, Pasture: 0.99, Grass: 0.99, No Person: 0.98, Mammal: 0.98

(B) No Person: 0.99, Water: 0.98, Beach: 0.97, Outdoors: 0.97, Seashore: 0.97

(C) No Person: 0.97, **Mammal: 0.96**, Water: 0.94, Beach: 0.94, Two: 0.94

Figure 1: Cow in different domains can't be correctly identified due to lack of causal feature usage within the learning model [8]

Therefore, there is a need to design ways to prevent any machine learning models from learning these spurious correlations, limiting their generalization capability. Since a model cannot generalize to any arbitrary unseen domains,

therefore an assumption has been made by the authors that we have an invariant predictor based on the stable causal features across domains.

Prior works like [11] propose an additional domain classifier trained from the representations learned by the feature extractor module. The network is then trained to minimize the label prediction loss and maximize the domain classification loss hence learning domain invariant representations. However, it has been seen that the domain invariant representations fail when the domain and the label are correlated.

We investigate the following claims from the original paper:

- *Claim 1*: The paper proposes an object invariant condition to estimate stable features to overcome the loopholes of the prior works.

- *Claim 2*: The paper proposes a novel 2-phase iterative algorithm to approximate the object-based matches.

## 3 Methodology

We make use of the code made available by the original authors for our study. Our major emphasis was to verify that the provided models and explanations stay true to the claims made in the paper. We further retrain their models on the provided dataset of Rotated-MNIST and Rotated Fashion-MNIST. We also attached our PyTorch implementation in the zipped folder to further verify the claims and experiment with the proposed extension.

### 3.1 Method descriptions

The problem statement that the paper is trying to solve is domain generalization, where we have access to data from multiple domains and distributions. The objective is to generalize to unseen domains at the testing phase. In order to overcome the flaws of the prior works, the authors in the paper further analyze whether the class conditional domain invariance objective is sufficient or not.

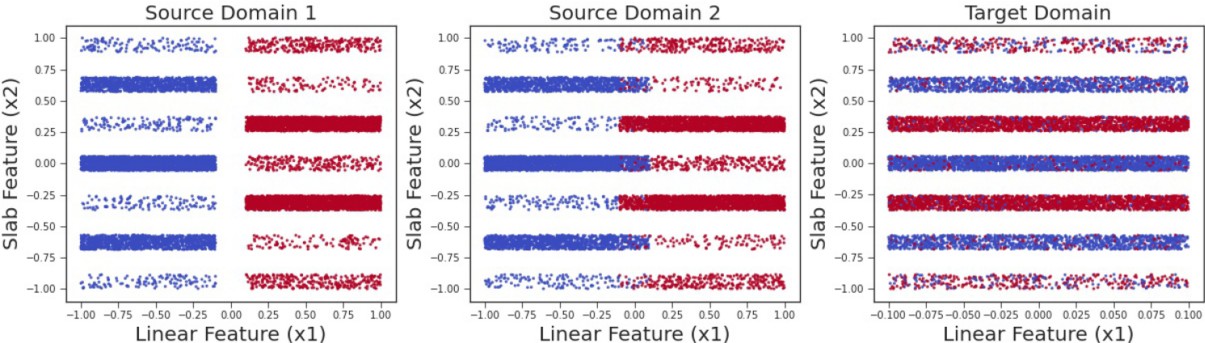

Figure 2: Slab Dataset (Slab (y-axis) is the stable feature) [21]

A simple counterexample has been shown in Figure 2, which illustrates a binary prediction task with two class labels on the slab dataset [1]. It has two types of features, where the first type of feature, X1, leads to a linear classifier separating the labels from the slab. The second feature, X2, leads to a more complex piecewise linear classifier separating the labels. The slab feature also has a small amount of noise represented by the low density of the opposite label. Overall, all the odd numbers slab correspond to the red colored points, and all the even numbers slab correspond to the blue colored points. The noise in the slab feature does not change across domains. On the other hand, the linear feature X1 has very low noise in the source domain, but it is completely noisy in the target domain. Due to the simplicity of the linear feature, a model might still learn the spurious linear feature over the stable slab feature.

One of the proposed methods is $perfmatch$. The method of perfmatch involves minimizing the loss $L$ across $m$-dimensions of the mapping function $h$ of the learnt representation of $X$ (denoted as $\Phi(X)$) to the output $Y$. The function also minimizes the distance between the learnt representations $\Phi()$ objects of the same class $j, k$ that exist in different domains $d, d'$ where the learnt matching $\Omega()$ of the same class objects $j, k$ $\Omega(j, k)$ is 1 for the different domains $d \neq d'$.

$$f_{perfmatch} = argmin_{h,\Phi} \sum_{d=1}^{m} L_d(h(\Phi(X)), Y) + \lambda \cdot \sum_{\Omega(j,k); d \neq d'} dist((\Phi(x_j^{(d)})), (\Phi(x_k^{(d')})))$$

The causal diagram in Figure 3 details the backdoor pass from object to domain, with the objects features separated into two categories, domain-dependent, and domain-independent. From the equation, the objective is to learn the correct Y for a given X, and this is achieved by using the domain-independent features $X_c$ to generalize across domains.

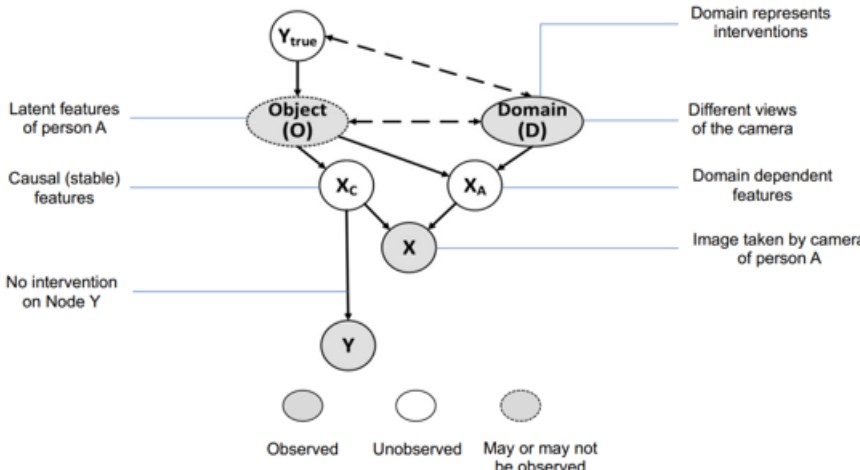

Figure 3: Causal graph proposed by the original authors [21] depicting the relation between $Y_{true}$, the causal features which are domain independent $X_c$ and Y

## 3.2 Datasets

The paper evaluated the matching-based methods on Rotated-MNIST and Fashion-MNIST, PACS, and Chest X-ray datasets.

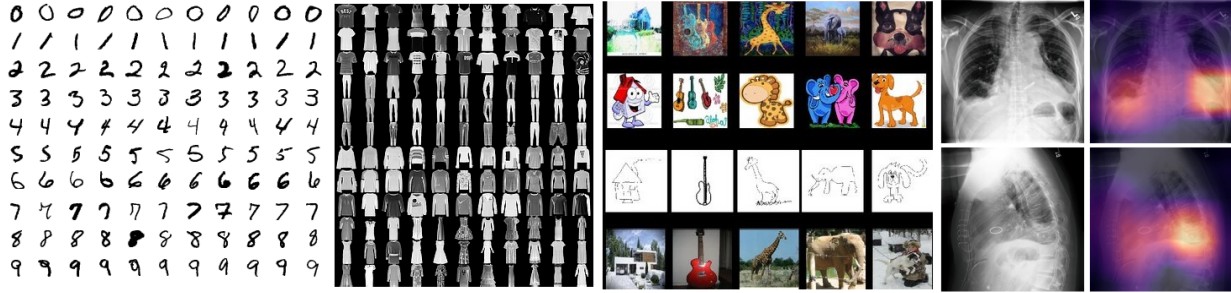

Figure 4: Datasets used: PACS [18, 28], CheXpert [3] MNIST [17] Fashion-MNIST[27]

***Rotated-MNIST and Fashion-MNIST***: It contain rotations of grayscale MNIST handwritten digits and fashion images from 0° to 90° with an interval of 15° [12]. Here, each angle represents a domain, and the task is to predict the class label. Following CSD, the paper reports the accuracy on 0° and 90° together as the test domain and the rest as the train domains.

***PACS dataset***: It contains a total of 9991 images from four domains: Photos (P), Art painting (A), Cartoon (C), and Sketch (S). The task is to classify objects over 7 classes. Inspired by [18, 28], the paper trains 4 models with each domain as the target using Resnet-18 [13], Resnet-50 [13], and Alexnet [16] network.

***Chest X-ray***: The paper introduces a harder real-world dataset based on Chest X-ray images from three different sources: NIH [26], ChexPert [15], and RSNA [4]. The objective is to identify patients with pneumonia. The original authors

inserted/produced a spurious correlation in the test domain by vertically translating class 0 in the training domains downwards, withholding the modification from the test domain.

### 3.3 Hyperparameters

We used hyperparameters stated in the original paper for most of our experiments. In cases where we deviated from the reported values, mostly due to computational resource and time limitations, we reported them in the discussion section. If a hyperparameter is not reported in the original paper, we either communicated with the authors to ask the hyperparameters or try out different values and report the result for all of them.

### 3.4 Experimental setup and code

We reran the code of the original authors on both public cloud infrastructure, such as Google Colab, and private GPUs that were available to us. We closely follow the experimental setup in the original paper for our experiments. Our scaling extension can be easily integrated with the source code and optimized similarly. Our implementations for all the experiments in this work are available in the supplementary material, and further support the reproducible research.

### 3.5 Computational requirements

We reran the code of the original authors on both public cloud infrastructures, such as Google Colab and private GPUs that were available to us. Google Colab provides a single 12GB NVIDIA Tesla K80 GPU that can continuously be used for up to 12 hours. We also ran the code locally on a 2 different machines. The first machine: The GPU in question is an Nvidia GeForce RTX 3080 10Gb GDDR6X. The CPU in this machine is an AMD Ryzen(TM) 7 5800 (8-Core, 36MB Total Cache, Max Boost Clock of 4.6GHz). The memory used was 32.0 GB DDR4 3466MHz, XMP. The second machine: i9-9900k, 1080Ti with 128 Gb DDR4 2666Mhz. We followed the setup in the original paper and implemented the network with the same number of iterations. Evaluating all the results with the saved models takes a good amount of time. It nearly took two days for some of the tables to generate the results. In conclusion, the code is not fast, but it can be run on a local machine. A GPU is heavily recommended because the code is slower without access to GPU.

## 4 Results

To reproduce the authors' experiments, we achieve approximately similar results to the original paper. We describe the results in the following sections:

### 4.0.1 Result 1

Table 1 presents an empirical analysis of various algorithms on the slab dataset to understand which invariance criteria can help to capture the stable (causal) features. The algorithms are evaluated based on the domain invariance and class conditional domain invariance criteria and experiment with the perfect match's new approach, which aims for domain invariance conditioned on the stable features. The results show that the perfect match approach does better than the domain invariance and class conditional domain invariance objective in learning stable features, emphasizing the need to choose the correct invariant criteria. The original authors made the observation that invariant representation learning by unconditional (DANN [11], MMD [19], CORAL [25]) and conditional distribution matching (CDANN [20], C-MMD [19], C-CORAL [25]), and matching same-class inputs (Random-Match [22]) have poor performance for the Target. We also observe this from our repeated experiment.

Table 1: Reproduced slab data-set results (Table 1 from original paper)

| Method | Source 1 | Source 2 | Target |
|---|---|---|---|
| ERM | **100 (0.00)** | 95.8 (0.27) | 64.6 (4.50) |
| DANN | 99.9 (0.07) | 95.1 (0.19) | 57.7 (1.69) |
| MMD | 99.9 (0.01) | 96.6 (0.25) | 70.3 (4.65) |
| CORAL | 99.9 (0.01) | 96.6 (0.25) | 70.3 (4.53) |
| RandMatch | **100 (0.0)** | 96.0 (0.30) | 66.9 (2.64) |
| C-DANN | 99.9 (0.07) | 95.1 (0.19) | 57.8 (1.69) |
| C-MMD | 99.9 (0.01) | 96.6 (0.25) | 66.8 (5.93) |
| C-CORAL | 99.9 (0.01) | 96.6 (0.25) | 70.2 (4.59) |
| PerfMatch | 99.8 (0.07) | **98.2 (0.21)** | **87.9 (4.26)** |

### 4.0.2 Result 2

Table 2 shows the replicated results for Rotated-MNIST & Rotated Fashion-MNIST for test domains 0° & 90°. MatchDG outperforms the comparison baselines for most of source distribution(CSD [24], MASF [10], IRM [6]). Source domains having the following angles (30, 45, 60) for Rotated Fashion-MNIST, MatchDG achieves an accuracy of 45.0%, and the next best method, CSD, achieves 38.9%.

Table 2: Rotated-MNIST and Rotated Fashion-MNIST to show various angles trained on vs accuracy achieved (Table 2 from original paper)

| Dataset | Source | ERM | MASF | CSD | IRM | Rand-Match | MatchDg | PerfMatch (Oracle) |
|---|---|---|---|---|---|---|---|---|
| Rotated MNIST | 15,30, 45,60, 75 | 92.0 (0.10) | 93.2 (0.20) | **94.9 (0.062)** | 92.5 (0.02) | 92.8 (0.28) | 94.5 (0.12) | 96.5 (0.51) |
| | 30,45, 60 | 75.7 (0.19) | 69.4 (1.32) | 80.4 (0.27) | 76.5 (1.73) | 79.6 (1.36) | **84.1 (0.43)** | 91.7 (0.50) |
| | 30,45 | 60.0 (1.84) | 60.8 (1.53) | 65.3 (0.87) | 60.2 (2.46) | 66.4 (1.46) | **72.0 (2.40)** | 80.4 (1.71) |
| Rotated Fashion MNIST | 15,30, 45,60, 75 | 78.1 (0.29) | 72.4 (2.90) | 81.4 (0.35) | 77.9 (0.72) | 77.0 (0.81) | **82.1 (0.23)** | 81.3 (0.22) |
| | 30,45, 60 | 37.3 (2.18) | 29.7 (1.73) | 38.9 (1.46) | 35.8 (1.38) | 35.7 (2.22) | **45.0 (1.57)** | 48.6 (3.90) |
| | 30,45 | 32.2 (0.47) | 22.8 (1.26) | 29.2 (0.54) | 32.1 (1.20) | 31.1 (0.53) | **35.0 (2.16)** | 37.7 (1.77) |

As of December 2021, the MatchDG algorithm holds the #1 ranking on the PapersWithCode website for Rotated Fashion-MNIST, with CSD as #2. The results we obtained for table 2 confirm that MatchDG performs better than the previous state-of-art technique CSD [24].

### 4.0.3 Result 3

Table 3 shows the repeated results whereby MatchDG outperforms ERM for overlap %. The table shows the advantage of PerfMatch for all 3 metrics over the default MatchDG variant for all metrics and each metric aligns with aligns with the other metrics for all baselines and models. This aligns with the results from the original authors as well.

Table 3: Overlap with perfect matches. top-10 overlap and the mean rank for perfect matches (Table 3 from original paper)

| Dataset | Method | Overlap (%) | Top 10 Overlap (%) | Mean Rank |
|---|---|---|---|---|
| MNIST | ERM | 14.3 (0.61) | 44.8 (2.57) | 32.1 (3.20) |
| | MatchDG (Default) | **27.9 (0.78)** | **63.5 (0.77)** | **18.6 (0.58)** |
| | MatchDG (PerfMatch) | 41.9 (5.48) | 80.5 (2.31) | 6.6 (0.36) |
| Fashion MNIST | ERM | 5.04 (0.09) | 20.9 (0.45) | 135.9 (2.91) |
| | MatchDG (Default) | **44.2 (2.62)** | **72.8 (1.48)** | **39.9 (7.36)** |
| | MatchDG (PerfMatch) | 68.0 (1.94) | 89.9 (1.50) | 9.5 (2.75) |

### 4.0.4 Result 4

Table 4 shows that for PACS dataset with ResNet-18 architecture, the results are competitive to the authors selected state of the art baselines (JiGen [9], DDAIG [29], SagNet [23], G2DM [5], CSD [24], RSC [14]) averaged over all domains. The MDGHybrid has the 3rd highest average, being beaten by DDEC [7] and RSC [14]. The paper reports MatchDG and MDGHybrid using a test domain validation, where MDGHybrid obtains comparable results to the best-performing baseline.

The authors original results for MatchDG also claim high rankings for the PACS [18, 28] dataset for both resnet18 and resnet50 on the PapersWithCode website. Our replicated results confirm these claims.

Table 4: Accuracy on PACS with Resnet 18 with test domain validation (Table 4 from original paper)

|  | P | A | C | S | Average |
|---|---|---|---|---|---|
| ERM | 95.38 | 77.68 | 78.98 | 74.75 | 81.70 |
| JiGen | 96.00 | 79.42 | 75.25 | 71.35 | 80.41 |
| G2DM | 93.75 | 77.78 | 75.54 | 77.58 | 81.16 |
| CSD | 94.10 | 78.90 | 75.80 | 76.70 | 81.40 |
| DDAIG | 95.30 | **84.20** | 78.10 | 74.70 | 83.10 |
| SagNet | 95.47 | 83.58 | 77.66 | 76.30 | 83.25 |
| DDEC | **96.93** | 83.01 | 79.39 | 78.62 | 84.46 |
| RSC | 95.99 | 83.43 | 80.31 | **80.85** | **85.15** |
| RandMatch | 93.07 | 77.03 | 76.82 | 75.59 | 80.63 |
| MatchDG | 96.17 | 79.09 | 79.15 | 75.88 | 82.57 |
| MDGHybrid | 96.43 | 81.84 | **80.52** | 76.50 | 83.82 |
| G2DM (Test) | 94.63 | 81.44 | 79.35 | 79.52 | 83.34 |
| RandMatch (Test) | 95.21 | 77.72 | 79.01 | 77.17 | 82.28 |
| MatchDG (Test) | **96.55** | 80.22 | 80.16 | **78.93** | 83.96 |
| MDGHybrid (Test) | 95.21 | **83.14** | **81.91** | 78.92 | **84.79** |

Table 5: Reproduced PACS resnet50 results (Table 5 from original paper)

|  | P | A | C | S | Average |
|---|---|---|---|---|---|
| DomainBed (ResNet50) | 97.80 | **88.10** | 77.90 | 79.10 | 85.70 |
| IRM (ResNet50) | 96.70 | 85.00 | 77.60 | 78.50 | 84.40 |
| CORAL (ResNet50) | 97.60 | 87.70 | 79.20 | 79.40 | 86.00 |
| RSC (ResNet50) | **97.92** | 87.89 | 82.16 | 83.35 | 87.83 |
| RandMatch (ResNet50) | 97.84 | 53.68 | 49.10 | 63.43 | 66.02 |
| MatchDG (ResNet50) | 96.71 | 83.98 | 82.45 | 80.74 | 85.97 |
| MDGHybrid (ResNet50) | 97.29 | 86.58 | **84.12** | **83.44** | **87.86** |

### 4.0.5 Result 5

Table 5 implement MatchDG on Resnet50 model used by the ERM in DomainBed. Adding MatchDG loss regularization improves the accuracy of DomainBed, from 84.79 to 87.86 with MDGHybrid. Also, MDGHybrid performs better than the prior approaches using Resnet50 architecture.

### 4.0.6 Result 6

Table 6 provides results for the Chest X-rays datasets from 3 different sources: RSNA, ChexPert and NIH. MDGHybrid outperforms other baselines for RSNA and Chexpert. However, on NIH MDGHybrid is outperformed by both ERM and CSD. The paper reasons these varying trends due to the inherent variability in "source domains, indicating the challenges of building domain generalization methods for real-world datasets". The original paper aligns generally with the replicated results but where MDGHybrid outperformed on Chexpert for our results, in their paper it under performed in the same manner that our results had an under-performance for NIH even though the original paper MDGHybrid attained the best result for NIH. Generally speaking the results hold.

Table 6: Chest x-ray results (Table 6 from original paper)

|  | RSNA | ChexPert | NIH |
|---|---|---|---|
| ERM | 59.4 (2.07) | 65.9 (0.93) | 61.6 (1.40) |
| IRM | 60.7 (2.87) | 66.1 (0.72) | 58.0 (1.44) |
| CSD | 65.7 (0.80) | 67.4 (0.90) | **63.6 (1.07)** |
| RandMatch | 60.5 (2.16) | 62.7 (2.93) | 59.3 (2.21) |
| MatchDG | 66.4 (2.19) | 65.3 (0.57) | 54.7 (2.18) |
| MDGHybrid | **76.7 (1.73)** | **67.6 (0.91)** | 61.4 (0.85) |

## 4.1 Results beyond original paper

In order to study the efficacy of the proposed method, we performed additional experiments by replicating their method in PyTorch for Rotated MNIST and Rotated Fashion MNIST. Our Pytorch implementation includes the entire method for train.py and data_gen_mnist.py under three settings. The first setting consists of the training samples for $0°$, $15°$, $30°$, $45°$, $60°$ and test samples for $75°$, $90°$. The second setting includes the training samples for $0°$, $15°$, $30°$, $45°$ and test samples for $60°$, $75°$, $90°$. Finally, the third setting includes the training samples for $45°$, $60°$, $75°$, $90°$ and testing samples for $0°$, $15°$, $30°$. Table 7 reports the accuracy for Rotated-MNIST and Rotated Fashion-MNIST datasets on target domains. For Rotated-MNIST, MatchDg outperformed all the other baselines; however, for Rotated-Fashion MNIST, we can see that CSD performs better, followed by MatchDg.

Table 7: Rotated-MNIST and Rotated-Fashion MNIST to show various angles trained on vs accuracy achieved (Table 2 from original paper, with out own modifications of angles)

| Dataset | Source | ERM | CSD | IRM | RandMatch | MatchDg | PerfMatch (Oracle) |
|---|---|---|---|---|---|---|---|
| Rotated MNIST | Train: 0,15,30,45,60 Test: 75,90 | 84.2 (1.05) | 89.1 (0.76) | 84.4 (1.84) | 87.5 (0.58) | **89.8 (0.54)** | 93.9 (0.29) |
| | Train: 0,15,30,45 Test: 60,75,90 | 73.0 (0.75) | 77.4 (0.51) | 72.8 (0.44) | 75.7 (1.23) | **78.5 (0.67)** | 88.9 (0.25) |
| | Train: 45,60,75,90 Test: 0,15,30 | 68.6 (1.00) | 73.5 (1.73) | 69.19 (1.14) | 72.1 (0.19) | **77.0 (0.74)** | 84.9 (0.72) |
| Rotated Fashion MNIST | Train: 0,15,30,45,60 Test: 75,90 | 53.0 (1.16) | **62.5 (1.28)** | 51.6 (1.42) | 53.8 (0.93) | 59.3 (0.71) | 65.5 (1.73) |
| | Train: 0,15,30,45 Test: 60,75,90 | 36.6 (0.48) | **43.7 (1.97)** | 36.0 (0.26) | 35.9 (1.02) | 41.7 (0.55) | 50.7 (0.27) |
| | Train: 45,60,75,90 Test: 0,15,30 | 30.8 (0.96) | **34.6 (0.72)** | 30.3 (1.16) | 30.4 (0.76) | 33.6 (0.43) | 36.7 (1.06) |

### 4.1.1 Additional Result 1

Table 8 contain the results for Rotated MNIST datasets using the LeNet architecture [19]. In this setup, there are six domains in total ($0°$, $15°$, $30°$, $45°$, $60°$, $75°$). The remaining five domains are used as source training domains for each test domain. Matching-based training methods RandMatch and MatchDG outperform prior work on all the domains.

Tables 9, 10, and 11 contain the results for appendix section results of DomainBed, fraction of perfect matches and overlap % when training on all domains.

### 4.1.2 Additional Result 2

The chars74k [2] dataset in Figure 5 offers an additional dataset to test the proposed algorithm in the paper. It contains characters A-Z, a-z, 0-9 from several domains, more specifically 64 classes (0-9, A-Z, a-z), 7705 characters obtained from natural images, 3410 hand-drawn characters using a tablet PC, 62992 synthesized characters from computer fonts. With the characters gathered from various sources, these sources can be considered in different domains. Thus, the algorithm should extract the causal features and be domain-independent, reflected in the results. Comparison to baselines should show it has an advantage. Unfortunately time did not allow for this testing, but it should be easy to see why this would be a good comparison for domain generalization.

Table 8: Accuracy for Rotated MNIST datasets using the LeNet architecture (Table 11 from original paper)

| Algorithm | 0 | 15 | 30 | 45 | 60 | 75 | Average |
|---|---|---|---|---|---|---|---|
| ERM | 90.1 (1.67) | 98.9 (0.26) | 98.0 (0.37) | 98.4 (0.22) | 97.9 (0.17) | 88.1 (1.25) | 95.23 |
| CCSA | 84.60 | 95.60 | 94.60 | 82.90 | 94.80 | 82.10 | 89.10 |
| D-MTAE | 82.50 | 96.30 | 93.40 | 78.60 | 94.20 | 80.50 | 87.60 |
| LabelGrad | 89.70 | 97.80 | 98.00 | 97.10 | 96.60 | 92.10 | 95.20 |
| DAN | 86.70 | 98.00 | 97.80 | 97.40 | 96.90 | 89.10 | 94.30 |
| CrossGrad | 88.30 | 98.60 | 98.00 | 97.70 | 97.70 | 91.40 | 95.30 |
| DIVA | 93.5 (0.3) | 99.3 (0.1) | 99.1 (0.1) | 99.2 (0.1) | 99.3 (0.1) | 93.0 (0.4) | 97.20 |
| RandMatch | **93.7 (1.10)** | **99.9 (0.05)** | **99.9 (0.12)** | **99.8 (0.12)** | **99.9 (0.05)** | 93.9 (0.45) | **97.84** |
| MatchDG | 93.6 (1.16) | 99.9 (0.12) | 99.7 (0.21) | 99.5 (0.12) | 99.8 (0.12) | **94.0 (0.57)** | 97.76 |
| PerfMatch | 96.2 (0.57) | 99.7 (0.05) | 99.7 (0.14) | 99.5 (0.12) | 99.6 (0.09) | 95.8 (0.49) | 98.42 |

Table 9: Accuracy for Rotated MNIST datasets using the DomainBed (Table 12 from original paper)

| Algorithm | 0 | 15 | 30 | 45 | 60 | 75 | Average |
|---|---|---|---|---|---|---|---|
| ERM | 95.4 (0.3) | 98.4 (0.1) | 98.4 (0.0) | 98.5 (0.1) | 98.2 (0.0) | 92.7 (1.3) | 96.93 |
| IRM | 95.9 (0.2) | 98.9 (0.0) | 99.0 (0.0) | 98.8 (0.1) | 98.9 (0.1) | 95.5 (0.3) | 97.90 |
| DRO | 95.9 (0.1) | 98.9 (0.0) | 99.0 (0.1) | 99.0 (0.0) | **99.0 (0.0)** | **96.9 (0.1)** | **98.10** |
| Mixup | 96.1 (0.2) | **99.1 (0.0)** | 98.9 (0.0) | 99.0 (0.0) | **99.0 (0.1)** | 96.6 (0.1) | **98.10** |
| MLDG | 95.9 (0.2) | 98.9 (0.1) | 99.0 (0.0) | **99.1 (0.0)** | **99.0 (0.0)** | 96.0 (0.2) | 98.00 |
| CORAL | 95.7 (0.2) | 99.0 (0.0) | **99.1 (0.1)** | **99.1 (0.0)** | **99.0 (0.0)** | 96.7 (0.2) | **98.10** |
| MMD | **96.6 (0.1)** | 98.9 (0.0) | 98.9 (0.1) | **99.1 (0.1)** | **99.0 (0.0)** | 96.2 (0.1) | **98.10** |
| DANN | 95.6 (0.3) | 98.9 (0.0) | 98.9 (0.0) | 99.0 (0.1) | 98.9 (0.0) | 95.9 (0.5) | 97.90 |
| C-DANN | 96.0 (0.5) | 98.8 (0.0) | 99.0 (0.1) | **99.1 (0.0)** | 98.9 (0.1) | 96.5 (0.3) | 98.00 |
| RandMatch | 95.3 (0.2) | 98.4 (0.03) | 98.1 (0.2) | 98.4 (0.2) | 98.2 (0.0) | 92.7 (0.4) | 96.91 |
| MatchDG | 95.7 (0.2) | 97.2 (0.1) | 98.8 (0.5) | 98.8 (0.2) | 98.9 (0.1) | 96.5 (0.3) | 97.65 |

Table 10: Accuracy results using a fraction of perfect matches during training (Table 13 from original paper)

| | MNIST | Fashion-MNIST |
|---|---|---|
| RandMatch | 92.8 (0.52) | 76.5 (0.13) |
| Approx 25% | 95.2 (0.12) | 77.3 (1.26) |
| Approx 50% | 95.5 (0.46) | 77.6 (1.34) |
| Approx 75% | 95.8 (0.21) | 79.1 (1.13) |
| PerfMatch 100% | 96.8 (0.32) | 82.5 (0.12) |

Table 11: Mean rank, Top-10 overlap, and overlap metrics for the matches learnt in the classification phase (Phase 2) (Table 14 from original paper)

| Dataset | Method | Overlap (%) | Top 10 Overlap (%) | Mean Rank |
|---|---|---|---|---|
| Rotated MNIST | RandMatch | 1.9 (0.14) | 11.7 (0.42) | 80.5 (0.99) |
| | MatchDG (Phase 2) | **15.7 (0.61)** | **42.86 (1.56)** | **40.12 (2.46)** |
| | PerfMatch (Oracle) | 71.3 (3.93) | 94.9 (1.09) | 2.0 (0.36) |
| Fashion MNIST (10k) | RandMatch | 1.6 (0.11) | 8.3 (0.38) | 291.1 (6.27) |
| | MatchDG (Phase 2) | **6.8 (1.77)** | **23.7 (4.65)** | **148.8 (26.07)** |
| | PerfMatch (Oracle) | 11.0 (1.00) | 35.2 (2.22) | 89.9 (8.68) |

## 5 Discussion

We observed several issues in the code; for example, in the dataset generation process, the authors randomly flipped the digits of the MNIST dataset during training, i.e., when they rotated a digit by 45°, it is not consistent with whether it will be clockwise or anticlockwise rotation. The issue was because they were using an inbuilt library of PyTorch, and because of that, when we modified the code to make the rotation consistent, the results improved. Also, for Table 1,

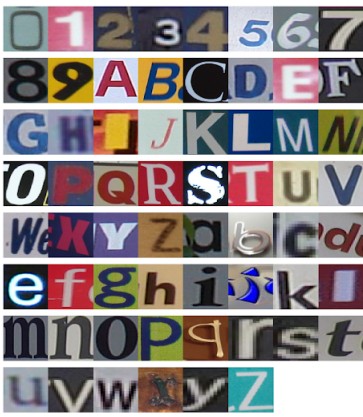

Figure 5: chars74k contains classes from multiple domains [2]

during code execution, we observed several errors and made necessary modifications. For instance, there were errors in the paths in slab_data.py. The same error was rectified by adding the correct path in the file: base_dir= os.getcwd() + '/data/datasets/slab/'. Secondly, during executing data_gen_syn.py for preparing slab dataset, datasets with spurr_list of 1.0 were not created. Therefore, in the file data_gen_syn.py we appended 1.0 i.e., the modified spur_corr_list is [0.0, 0.10, 0.20, 0.90, 1.0]. On Windows machines, a freeze_support() error was encountered, and thus train.py and test.py needed to have the main() method added (problem is specific to windows only, believed to be an underlying issue with python). Some basic installations were needed for the libraries like torchcsprng and opacus.

## 5.1 What was easy

The official GitHub page of the paper has the authors' open source code, which was helpful. The experiments described in the paper were done on widely-used standard datasets. Therefore, implementing each experiment was relatively easy to do. Furthermore, since many of the parameters were reported in the scripts, we did not need much tuning in most experiments.

## 5.2 What was difficult

Though implementing each experiment is relatively simple, the numerosity of experiments proved to be challenging. In particular, each experiment in the original setting requires training a network for a significant number of iterations. We sometimes changed the settings in these cases. However, these changes did not affect the interpretability of the final results.

## 5.3 Communication with original authors

We emailed the authors and received prompt responses to our questions regarding the provided Jupyter reproduction notebooks. Some tables had multiple runs for the same technique, but it was unclear how to execute the alternative runs. For reproducing Table 1 in the original paper, it was unclear how we could obtain quantitative values for source 1, source 2, and target. As per the script, it was producing values for source and target. Therefore, We communicated with the authors via email and asked them to explain the condition used in the experiments more clearly. They stated that the numbers obtained are evaluated on the target domain/test dataset under different validation strategies. Hence we cannot break them down into source 1 and source 2. Running the script with the evaluate flag would evaluate the trained model and provide per domain accuracy (source 1, source 2).

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
