# OpenReview forum: "[Re] Domain Generalization using Causal Matching"
_ML_Reproducibility_Challenge/2021/Fall — RC2021_

### Official Review · Reviewer_jzi3 · 2022-02-25
**Successful reproduction and application to new data sets**

**Rating:** 7
**Confidence:** 4

**Review:**

The reproducibility report for "Domain Generalization using Causal Matching" clearly states the problem statement of the original paper: learning to generalize in order to deal with domains unseen in the training data set. Source code for the experiments in the original paper was available and used for the reproducibility study. The authors contacted the original authors successfully.

All experiments in the original paper have been reproduced successfully. Small deviations are explained well. The authors performed additional experiments using different rotation angles as training and test domains as well as an extra data set. For the reproduction the original hyperparameters were used and no ablation study was performed.

The report is well written and discusses the results of the reproducibility study at length. Additional experiments show that the original algorithm also works on data sets not tested before which nicely fits the topic.

---

### Official Review · Reviewer_bm4j · 2022-03-06
**Most of the results reproduced with new understandings**

**Rating:** 5
**Confidence:** 3

**Review:**

1. In practice, this report reproduces almost all the results in the original paper. However, the author failed to clearly and concisely state the scope of reproducibility. I did see such a scope statement in the first paragraph.

2. The author uses public infrastructures and the original author's code without providing the exact codebase they used.

3. The report shows fair communication with original authors. However, the result of communication is not perfectly satisfying. The exact experiment settings remain unclear.

4. The report uses the same hyperparameters settings as the original paper. They find several errors in the original implementation and improve the results of the experiment.

5. The report claims that the hardness of reproducing results is that "each experiment in the original setting requires training a network for a significant number of iterations". However, the report does not show how the number of iterations affects the results and how many iterations are required to reproduce the result.

---

### Meta-Review · Area_Chair_vkX7 · 2022-04-09

**Recommendation:** Accept
**Confidence:** 4

**Metareview:**

Reviewers praised that the authors reproduced many experiments from the original paper, and even found several errors in the original implementation.

---

### Decision · Program_Chairs · 2022-04-09

**Decision:**

Accept

**Comment:**

Following the recommendation of reviewers and meta-reviewer, the paper is accepted for ML Reproducibility Challenge 2021, and will be published in the upcoming special edition of ReScience Journal.